# Air Layering Improves Rooting in Tree Peony Cultivars from the Jiangnan Group

Ying Zhang, Shui-Yan Yu *  and Yong-Hong Hu *

Shanghai Key Laboratory of Plant Functional Genomics and Resources, Shanghai Chenshan Botanical Garden, Shanghai 201602, China
* Correspondence: yushuiyan1982@163.com (S.-Y.Y.); huyonghong@csnbgsh.cn (Y.-H.H.)

**Abstract:** Tree peony (*Paeonia suffruticosa* Andr.), a unique traditional flower in China, is famous for its ornamental value, medical use, and edible oil production. Traditional propagation methods, such as sowing, dividing, and grafting do not allow the large-scale production of selected peony varieties. Therefore, the objective of our study is to evaluate an air-layering technique on the rooting success of three tree peony cultivars ('Baoqing Hong', 'Quehao', and 'Xishi'). The experiments were established through consideration of the influence of the time of the year the rooting was performed (mid-May, mid-June, or mid-July) and the growth regulators (1-naphthaleneacetic acid-NAA and indole-3-butyric acid-IBA) applied at different concentrations (1000 mg/L, 1500 mg/L, 2000 mg/L). The results showed that the rooting rate was the highest when the air-layering time occurred in mid-June, and the rooting rate of 'Quehao' was found to be the most significant, reaching 100%. The rooting percentages of 82.86% and 77.14% were obtained for 'Baoqing Hong' and 'Xishi', respectively. The growth regulators affected the rooting performance of the three cultivars differently. The rooting parameters of 'Quehao' were negatively correlated with the concentration of NAA but positively correlated with IBA, whereas the 'Baoqing Hong' and 'Xishi' cultivars showed no dose dependence for the supplied growth regulators. Root number, root tip number, and maximum root length in 'Quehao' were higher than those of the other two cultivars. The conclusion of our study is that the air-layering technique is a suitable method for achieving satisfactory propagation of selected tree peony cultivars.

**Keywords:** propagation; tree peony; air layering; rooting parameters; growth regulators



## 1. Introduction

Tree peony (*Paeonia suffruticosa* Andr.) is a traditional flower that is unique to China. It is a rare ornamental flower as well as a medicinal and food plant. Its roots have hypoglycemic, antibacterial, anti-inflammatory, anti-atherosclerosis, anti-arrhythmia, anti-convulsion, liver-protection, immunity-enhancing, and other pharmacological effects [1,2]. Tree peony seed oil contains a large amount of unsaturated fatty acids, such as $\alpha$-linolenic acid ($\omega$-3 family), linoleic acid, and oleic acid, which have anti-inflammatory, anti-tumor, lipid-lowering, and immunity-improvement effects [3,4]. In addition, tree peony stamens can be used to prepare tea, and the petals can be processed and refined into essential oils that are widely used in daily health care and various industries [5,6]. With the development of deep-processing products, the tree peony industry has gradually developed into a comprehensive and multi-product industry.

Increased market demand has promoted the rapid development of the tree peony seedling industry, but the capacity for domestic tree peony seedling production cannot meet market demands. This is because of the long life cycle of peony, which takes 4–6 years from sowing to flowering and fruiting [7]. This biological characteristic severely restricts peony industrialization. The traditional propagation methods for tree peony include sowing, dividing, and grafting, which have low propagation coefficients and long seedling

periods, as well as inconsistent quality [8]. Although tissue culture could represent a valid method for vegetatively propagating tree peony, several drawbacks are still to be addressed with this method, including poor rooting, vitrification, a low multiplication rate, difficult acclimatization, and a low in vivo survival rate [9–18]. Large-scale seedling production is based on grafting and dividing methods, but several issues limit breeding and industrial development [19–21]. The main factors affecting the success of grafting include grafting time, rootstock, scion, and grafting method. The grafting time for tree peony is generally October–November, with graft survival being limited at other times. Even when plant grafting is carried out at the optimal time, diseases and insect pests may occur, and sudden low temperatures in early spring and other environmental factors can inhibit survival. The grafting survival rate of various peony varieties in the second year is only 50%–85% [22].

Tree peonies with high ornamental value are mostly seedless and cannot be sexually reproduced. The roots of *P. lactiflora* and *P. ostii* are typically selected as the rootstock. Due to the characteristics of the rootstock, the branches of the rootstock are often budded from the root, affecting the growth of the grafted seedlings [8]. The grafting affinity of some varieties is also weak, affecting the grafting success. Although some tree peony varieties can be sexually reproduced, the growth cycle is very long. It takes 5–6 years from sowing to flowering plants and production of seeds. Thus, the seed yield is low, and the plants cannot be quickly placed into the market [7]. The characteristics of the offspring of superior plants propagated through grafting are greatly affected by the rootstock, and the screened high-quality tree peony plants cannot be placed into the market quickly and efficiently, thereby hindering the development of the peony industry.

Air-layering propagation technology can retain the favorable traits of the mother plants as well as encourage early flowering and fruiting [23]. This method has been applied to the propagation of different plants with low propagation coefficients and has achieved good results, such as in *Phoebe bournei* [24], *Camellia japonica* [25], and *Saraca* 'Siji Flower' [26]. The progeny obtained by air layering have high purity and excellent traits, are not influenced by the rootstock, and exhibit early flowering and fruiting. The seedlings obtained by air layering grow better in the second year, at which point they can be placed on the market. Although the air-layering propagation technique is widely used, no relevant studies on this topic can be found for tree peony or Paeonia (data source: Web of Science). Therefore, it is a novel approach to adopt an air-layering technique for tree peony propagation. The purpose of this study is to explore whether the air-layering technique can be successfully implemented in tree peony to establish an effective propagation method that is different from the traditional propagation method.

## 2. Materials and Methods

### 2.1. Plant Material

Three varieties (Baoqing Hong', 'Quehao', and 'Xishi') belonging to the Jiangnan group were selected from plants previously grown at the Luoyang Shenzhou Peony Garden, and were subsequently transferred to the Shanghai Chenshan Botanical Garden (31°4′52″ N, 121°10′14″ E). The plants have been kept under the same environmental and cultivation conditions since 2014, and currently there are hundreds of living plants growing in the tree peony planting base of the Shanghai Chenshan Botanical Garden. The present experiment was conducted from May to November in 2020 with plants that were 8–10 years old.

### 2.2. Air-Layering Procedure

On the mother plant, an area about 1 cm away from the end node of two-year-old branches was selected, and the phloem was peeled off completely to expose the xylem. The ring-stripping incision was about 1 cm. A cotton swab was used to apply the rooting treatment by dipping into the incised surface. The center of a 20 × 20 cm square film was cut to the midpoint of one side and used as a funnel. The film funnel was filled with soaking medium, which consisted of 50% moss and 50% peat (Klasmann-Deilmann 413, pH 5.5–6.5). The moss was cut appropriately in advance, and the length was controlled

within 1 cm. After filling the soaking medium, the top of the film was secured with a rust-proof black tie wire so that it was easy to naturally collect rain. If it did not rain for a long time, water was properly supplied to keep the medium moist. In late September or October, when the tree peonies were ready for transplanting, the branches below the film opening were removed and the rooted seedlings were transplanted into the flowerpot. The seedlings were then propagated using standard procedures [27].

### 2.3. Influence of the Time of the Year on Air Layering

The experiment was performed with the three already cited tree peony varieties and three periods for air layering were considered: mid-May in Spring, and mid-June and mid-July in Summer. The recorded climate conditions during our experiment were:

a.      average temperature: 16–24 °C in May, 21–27 °C in June, and 25–32 °C in July
b.      average precipitation: 112 mm in May, 169 mm in June, and 151 mm in July

For each period, seven individual mother plants replicates were considered, and 5 fresh branches of the current year were selected from each plant for the air layering. Then, 1000 mg/L of an equal ratio of 1-naphthaleneacetic acid (NAA) and indole-3-butyric acid (IBA) was applied to all ring strips.

### 2.4. Influence of Growth Regulators on Air Layering

The experiment was performed in mid-June 2020, and the three cultivars already mentioned were considered. In accordance with previous studies on air layering and with experience in tree peony cultivation, NAA and IBA were used as rooting growth regulators at concentration 1000 mg/L, 1500 mg/L, and 2000 mg/L [28–30]. A control treatment (no growth regulators applied) was considered too. For each cultivar, 3 mother plants and 5 branches for each mother plant were considered, resulting in a total of 105 treatments for each tested cultivar.

### 2.5. Evaluation of the Air-Layering Propagation in Tree Peony

In October (a suitable season for transplanting tree peony), 3–4 months after having applied air layering, the branches were cut below the membrane opening. After removing the tie wire, film, and the medium, the relevant growth parameters were measured.

Determination of air-layering efficiency: each air layer was classified according to the rooting efficiency, as follows: (i) air layers with more than 10 growing first-order roots (referred to as 1) and (ii) air layers with fewer than 10 growing first-order roots or with no roots (referred to as 0). According to this classification, for each plant, a 'rooting number' was derived, and then the air-layering efficiency was calculated as follows:

$$\mathrm{Air-layering\ efficiency} = (\mathrm{Total\ 'rooting\ number'}\ /\ \mathrm{Total\ air\ layers}) \times 100\% \quad (1)$$

In order to highlight the root architecture that influenced plant growth and stability, the following parameters were taken into account:

a.      Root system quantity, which was evaluated through the number of taproots and root tips for each air layer. This was considered an indication of the rooting efficiency
b.      Root system quality, which was determined by considering the primary root length. The taproots were determined one by one by measuring the length from the base to the top with a ruler, accurate to 1 mm. The sum of all primary root lengths of a single air layering represented the total length of the primary roots of the air layering, and the longest primary root length indicated the maximum primary root length of the air layering.

### 2.6. Data Processing

Excel 2017 (Microsoft Corp., Redmond, MA, USA), GraphPad Prism 7.0 (GraphPad Software, San Diego, CA, USA), and Adobe Illustrator CS6 (Adobe Mercury, San Jose, CA, USA) were used for experimental data processing and mapping. Standard deviation

analysis was performed using STDEV in Excel 2017, and a variance analysis was performed using ANOVA analysis ($p < 0.05$) in GraphPad Prism 7.0.

## 3. Results

### 3.1. Influence of the Time of the Year on Air Layering

Figure 1 shows the air-layering procedure in tree peony roots and the growth of the air layers. A one-centimeter ring-stripping incision is shown in Figure 1A. After growth regulators were applied and a thin film was placed on the ring-stripped surface (Figure 1B), callus formation began on the ring-stripped branches in the upper air layer at about 15 d, rooting began at 25–30 d, and white roots filled the matrix at 60–70 d (Figure 1C,D). Figure 1E,F shows the air layer growth in pots one year from the air layering.

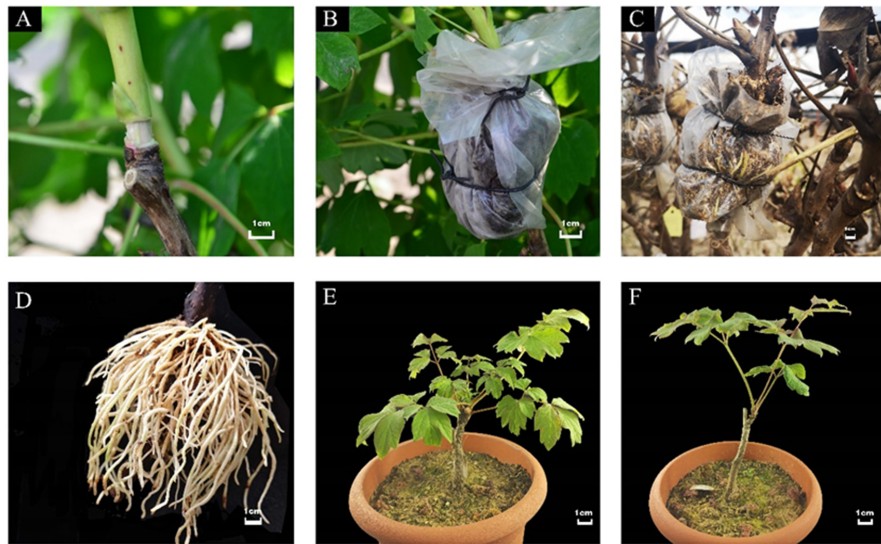

**Figure 1.** Steps in air layering of tree peony. (**A**) One-centimeter ring-stripping incision. (**B**) Matrix dressing. (**C**) The substrate is covered with new roots. (**D**) The new roots under air layering. (**E**,**F**) Air-layer growth in pots one year from the air layering.

The number of first-order roots of each layer was counted and calculated according to the formula already stated for air-layering efficiency. Table 1 highlights that the best results were reached when the air layering was carried out in mid-June, and this was true for all the three tested varieties, although a different performance could be scored (the air-layering efficiency was 82.86% ± 10.1, 100% ± 0% and 77.14% ± 8.9 for 'Baoqing Hong', 'Quehao' and 'Xishi' respectively). Generally speaking, the worst results were obtained when the air layering was performed in mid-July, except for the variety 'Baoqing Hong', for which the efficiency of the air layering was similar when the procedure was performed in May or July.

**Table 1.** Air-layering efficiency for the three tested tree peonies.

| Cultivars | Time | Total Air Layers | Rooting Number | Air-Layering Efficiency (100%) |
|---|---|---|---|---|
| 'Baoqing Hong' | mid-May | 35 | 27 | 77.14 ± 9.1 |
| 'Baoqing Hong' | mid-June | 35 | 29 | 82.86 ± 10.1 [a] |
| 'Baoqing Hong' | mid-July | 35 | 27 | 77.14 ± 9.3 |
| Quehao | mid-May | 35 | 33 | 94.29 ± 3.8 [b] |
| Quehao | mid-June | 35 | 35 | 100 ± 0 [b] |
| Quehao | mid-July | 35 | 30 | 85.71 ± 10.3 [a] |
| Xishi | mid-May | 35 | 26 | 74.29 ± 9.8 |
| Xishi | mid-June | 35 | 27 | 77.14 ± 8.9 |
| Xishi | mid-July | 35 | 22 | 62.86 ± 12.1 |

Note: Air layering was applied over three different periods, and a solution of NAA + IBA (1000 mg/L each) was applied to stimulate the rooting. Data are mean average values ± standard error. [a] indicates significant differences, $0.01 < p < 0.05$; and [b] indicates highly significant differences, $0.01 < p < 0.05$.

### 3.2. Influence of Growth Regulator on Air Layering

As indicated in Figure 2A, the number of taproots in 'Baoqing Hong' under NAA treatment was twice as high as in the control (CK). Under IBA treatment, the number of taproots was slightly higher than that of the control group, though the difference was not significant. The number of root tips of 'Baoqing Hong' was lower than that of the control group under the two growth-regulator treatments (Figure 2A). Figure 2B indicates that the total length and maximum length of the primary roots of 'Baoqing Hong' under NAA treatment were similar to those of the control group. When IBA was applied, the total and maximum root length were slightly lower than those of the control group. Figure 2C showed that the numbers of primary roots and root tips of 'Quehao' with the growth regulator were significantly higher than those of the control (CK), which under the NAA treatment were slightly higher than those under the IBA treatment. As indicated in Figure 2D, the total length and maximum root length of the primary roots in 'Quehao' with the growth regulator were significantly higher than those of the control (CK), which under the IBA treatment were similar to those under the NAA treatment. As shown in Figure 2E,F, the number of main roots in the root system of 'Xishi' under the NAA and IBA treatments was significantly higher than that of the control group. The number of root tips under the NAA treatment was slightly higher than that of the control group, but significantly higher under IBA treatment. The total length and maximum length of the primary roots were slightly higher under NAA treatment than in the control (CK), but lower under IBA treatment. The results showed that NAA and IBA could affect the number and length of roots differently, although a varied response to the treatments was noted.

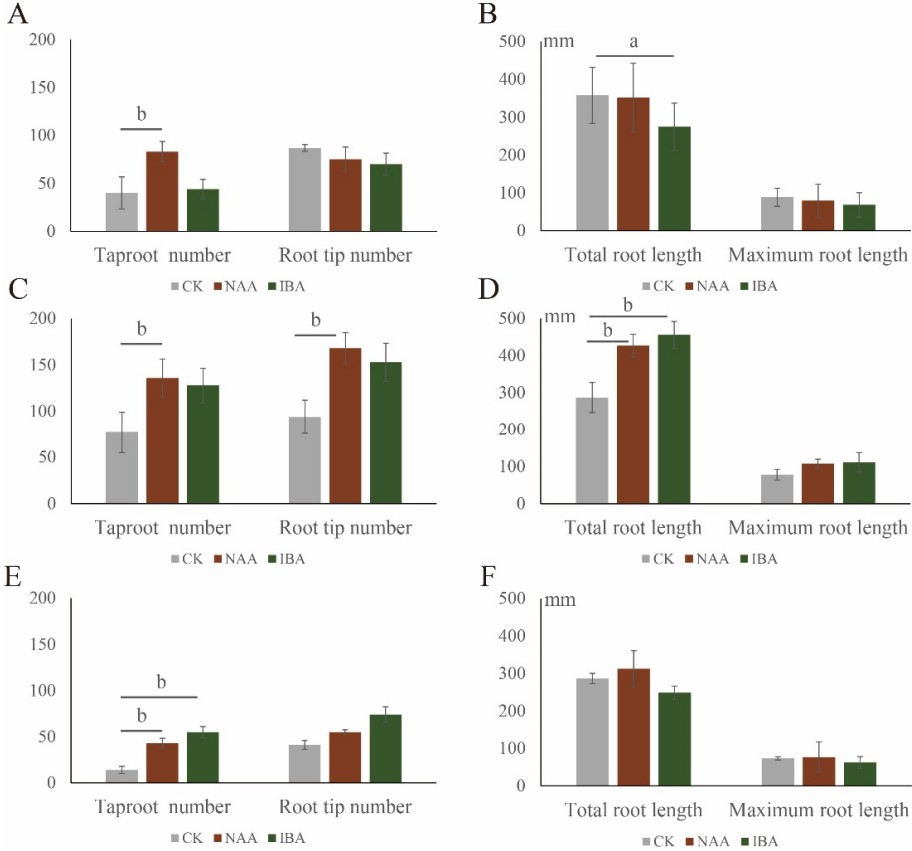

**Figure 2.** Rooting efficiency for air layering of tree peony when different growth regulators (IBA and NAA) were supplied at the concentration 1000 mg/L. The evaluation was carried out for three tree peony varieties: 'Baoqing Hong' (**A,B**), Quehao (**C,D**) and Xishi (**E,F**). A control treatment (no growth regulators applied = CK) was considered for each variety. [a] indicates significant differences, $0.01 < p < 0.05$; and [b] indicates highly significant differences, $0.01 < p < 0.05$.

### 3.3. Influence of Different Concentrations of Growth Regulators on Air Layering

In a subsequent experiment, three concentrations of the growth regulators NAA and IBA (1000 mg/L, 1500 mg/L and 2000 mg/L) were applied in the air layering carried out in mid-June. Table 2 shows that growth regulator choice and the concentration applied could greatly affect the root architecture; also in this case, we have to highlight that there was a genotype-dependent response.

**Table 2.** Air-layered root traits of the tree peony varieties under the different treatments.

| Cultivars | Treat Agents | Concentration (mg/L) | Taproot Number | Root Tip Number | Total Root Length (mm) | Maximum Root Length (mm) |
|---|---|---|---|---|---|---|
| 'Baoqing Hong' | CK | 0 | 40 ± 6.2 | 101 ± 6.5 | 357.7 ± 11.9 | 88.6 ± 6.7 |
| | NAA | 1000 | 83 ± 10.7 [b] | 75 ± 12.9 | 335.1 ± 31.1 | 79.2 ± 12.6 |
| | | 1500 | 62 ± 5.1 [a] | 84 ± 2.1 | 250.7 ± 26.6[a] | 60.3 ± 4.1 |
| | | 2000 | 64 ± 13.1 [a] | 87 ± 4.3 | 282.4 ± 26.1 [a] | 67.5 ± 5.5 |
| | IBA | 1000 | 44 ± 4.6 | 70 ± 2.3 | 274.8 ± 22.1 [a] | 71.6 ± 4.5 |
| | | 1500 | 114 ± 12 [a] | 155 ± 10.5 [a] | 324.5 ± 27.7 | 72.5 ± 6.1 |
| | | 2000 | 45 ± 7.5 | 67 ± 6.1 | 340.4 ± 11.7 | 78.7 ± 7.6 |
| 'Quehao' | CK | 0 | 77 ± 5.1 | 94 ± 18.2 | 364.7 ± 21.1 | 77.7 ± 5.6 |
| | NAA | 1000 | 136 ± 17.1 [a] | 168 ± 8.9 [a] | 426.8 ± 30.4 [a] | 107.3 ± 12.6 [a] |
| | | 1500 | 110 ± 13.5 [a] | 138 ± 3 | 396.4 ± 22.2 | 91.5 ± 9.6 |
| | | 2000 | 69 ± 2.5 | 79 ± 7.5 | 312.9 ± 24.6 | 72.7 ± 3.5 |
| | IBA | 1000 | 128 ± 7.8 [a] | 153 ± 28.2 [a] | 455.9 ± 38.9 [a] | 110.9 ± 8.3 [a] |
| | | 1500 | 204 ± 3.8 [b] | 198 ± 5.2 | 471.3 ± 24.5 | 111.3 ± 3.4 [a] |
| | | 2000 | 222 ± 5.1 [b] | 255 ± 23.8 [b] | 564.8 ± 38.2 [b] | 120.6 ± 18.9 [b] |
| 'Xishi' | CK | 0 | 21 ± 0.6 | 41 ± 3 | 286.3 ± 13.9 | 73.5 ± 4.1 |
| | NAA | 1000 | 43 ± 5.3 [b] | 55 ± 2.5 | 312.2 ± 5.1 [a] | 76.5 ± 3.1 |
| | | 1500 | 37 ± 1.5 | 42 ± 2 | 167.4 ± 16.2 [a] | 43.3 ± 2.2 |
| | | 2000 | 31 ± 10.3 | 41 ± 1.7 | 225.6 ± 11.8 | 60.3 ± 3.7 |
| | IBA | 1000 | 55 ± 6.1 [b] | 74 ± 3.6 | 248.5 ± 18.4 | 62.2 ± 7.1 |
| | | 1500 | 43 ± 2.5 [b] | 52 ± 5.1 | 263.9 ± 18.8 | 71.63 ± 8.6 |
| | | 2000 | 55 ± 0.6 [b] | 64 ± 4.9 [a] | 278.5 ± 25.9 | 63.4 ± 4.6 |

Note: Data are the mean average values ± standard errors. [a] indicates significant differences, $0.01 < p < 0.05$; and [b] indicates highly significant differences, $0.01 < p < 0.05$. CK, blank control; NAA, 1-naphthaleneacetic acid; IBA, indole-3-butyric acid.

Generally speaking, the taproot number was enhanced when a growth regulator was applied. In the case of 'Baoqing Hong' variety, the highest number of taproots (114 ± 12) was obtained when the growth regulator IBA was applied at 1500 mg/L. Furthermore, with this IBA concentration the highest number of root tip was obtained (155 ± 10.5). Satisfactory results for all the root parameters were obtained in 'Quehao' variety at IBA 1500 or 2000 mg/L, the latter concentration being the best (taproots number: 222 ± 5.1; root tip number: 255 ± 23.8; total root length: 564.8 ± 38.2 mm and maximum root length 120 ± 18.9 mm). Interestingly, for this variety, we found that when the NAA concentration was increased, the root parameters decreased (particularly the number of taproots and root tips). On the contrary, decreasing the IBA concentration from 2000–1000 mg/L, the number of taproots was reduced by 40%. (Table 2). In the case of 'Xishi' variety, the lowest number of taproots were always obtained compared to the other varieties; the best results were obtained when IBA was provided at 1000 mg/L or 2000 mg/L (number of taproots: 55 ± 6.1 and 55 ± 0.6 respectively). For this variety, the root parameters showed an independence from the growth-regulator concentration applied (Table 2).

### 3.4. Evaluation of the Air-Layering Propagation in the Various Tree Peony Cultivars

The rooting characteristics of the three tree peony cultivars 'Baoqing Hong', Quehao', and 'Xishi' in response to the NAA and IBA treatments were compared at the same concentration of 1000 mg/L. The results in Figure 3 show that 'Quehao' had a significantly higher number of taproots and root tips, root lengths, and maximum root lengths than

'Baoqing Hong' and 'Xishi', exhibiting highly advantageous root growth under air layering (Figure 3A–D). Under NAA treatment, the root parameters of 'Baoqinghong', particularly the number of taproots and the total number of root tips, were significantly higher than those of 'Xishi'. However, there was no significant difference in the total length and maximum root length of the primary roots between the two cultivars. The root parameters (taproots number, root tip number, total root length and maximum root length) between 'Baoqing Hong' and 'Xishi' were similar under IBA treatment.

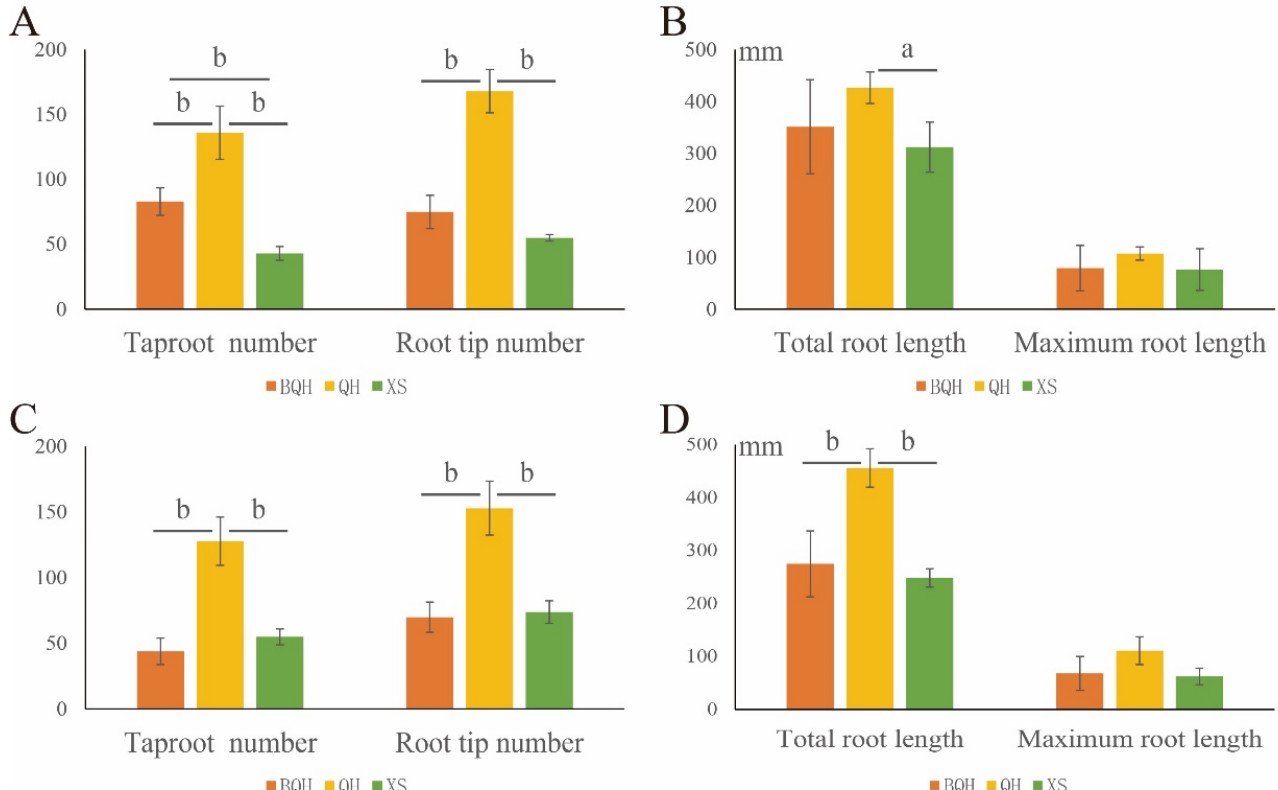

**Figure 3.** Comparison of the root characteristics among the three cultivars of tree peonies under air layering. (**A**) Comparison of the number of taproots and root tips of three cultivars treated with NAA at 1000 mg/L; (**B**) the total length and maximum root length of the primary roots of three cultivars treated with NAA at 1000 mg/L; and (**C**,**D**) comparison of the same root characteristics among the three cultivars treated with IBA at 1000 mg/L. BQH: Baoqing Hong; QH: Quehao; XS: Xishi. [a] indicates significant differences, $0.01 < p < 0.05$; and [b] indicates highly significant differences, $0.01 < p < 0.05$.

## 4. Discussion

Tree peonies prefer dry and cold growth conditions, while the humid and hot climate in Jiangnan area of China is not conducive to the growth of tree peony. Some varieties of tree peony have better growth adaptability, but the performance of different varieties differs greatly [7,31]. The three tree peony cultivars, 'Baoqing Hong', 'Quehao', and 'Xishi', belong to the traditional Jiangnan cultivar group, and they have good growth adaptability in the Jiangnan area [7,31]. To avoid poor experimental data due to the bad growth, these three cultivars were selected to undergo air-layering propagation experiments. In our experiment designed to evaluate the air-layering method for propagating the tested tree peony varieties, we found that the variety 'Quehao' performed better than the varieties 'Baoqing Hong' and 'Xishi'. The air-layering efficiency for this variety was good, and the root features were satisfactory, showing a dose dependence in response to the growth regulator treatment. This could be ascribed to the existence of root primordia in the cortexes of the branches of the tree peonies, the number of which varies by cultivar [7]. 'Quehao' belongs to the Huizi series of the Jiangnan cultivar group, which has a long cultivation history and better

growth adaptability than 'Baoqing Hong' and 'Xishi' [7,31]. Considering the times of the air layering and flowering, it is speculated that the differences in rooting after air layering may have been related to the early and late flowering times among cultivar varieties.

The three air-layering times chosen were mid-May at the end of spring, mid-June in early summer, and mid-July in midsummer, during tree peony fruit development. The air-layering experiment results showed that air-layering time had a significant effect on the rooting rate among the three tree peony cultivars. Similar findings have also been reported in guava [32,33] and pomegranate cv. *Bhagwa* [34]. The rooting rates of the three tree peony cultivars were significantly higher when layered in early summer than in spring and midsummer. In early summer (mid-June), the seeds of tree peonies in the Jiangnan region are in the period of nutrient accumulation [35], and therefore, the plant needs to be supplied with a large quantity of nutrients for the seeds. It is possible that the phloem of the branches is more active during this time, and this would account for the rapid rooting on the callus. To corroborate this statement, it is reported in the literature that air layering in *Ilex verticillate* provided the best results in June, and this was ascribed to the fact that the fruits were forming and the branches were growing rapidly in that period [36]. Furthermore, in the Shanghai area, mid-June is the Meiyu season [37]. In our experiment, the rooting of the air layers started 25–30 days after the air layering, and this happened precisely in the Meiyu season. Humidity and temperature (22–25 °C) were very suitable for air-layered branch rooting and the promotion of root growth. This result was consistent with other studies, which found that air layering was beneficial during the wet season when there was high humidity [29,30]. It has been revealed that the rooting rate of layered shoots in the shea tree (*Vitellaria paradoxa*) under wet conditions is higher than that under dry conditions [29]. The air-layered rooting percentage in guava (*Psidium guajava*) is due to a particular combination of humidity, rainfall, and temperature [30], and similar findings have also been reported by Shrivastava in *Puncia granatum* [38], Sharma and Grewal in lithci [39], and Sarker and Ghose in guava [40].

The growth regulators NAA and IBA have a great influence on rooting characteristics [41,42]. The rooting features of 'Quehao' under air layering were negatively correlated with the concentration of NAA at 1000–2000 mg/L and positively correlated with the concentration of IBA at 1000–2000 mg/L. These correlations were not observed for the other tested varieties. It has been observed that IBA had a better rooting effect than NAA in air-layered rooting in 'Quehao'. According to our findings, the air-layered rooting characteristics had a significant difference between the treatment types and concentrations among the three cultivars. In air-layering studies of *Lasiococca comberi*, the rooting effect of IBA has been reported to be much higher than that of NAA and indole-3-acetic acid (IAA) [43]. In contrast, in a study of *Phoebe bournei* using air layering, the rooting percentage was significantly higher under the NAA treatment than that of IBA and IAA [24]. Therefore, NAA as a treatment agent has a less clear effect on layering rooting than the IBA treatment. The concentration of IBA used as the air-layering rooting growth regulator is obviously different among plant species. In the case of propagation by air layering in avocado (*Persea* sp.), the application of IBA increased the rooting percentage to a maximum of 74% at a concentration of 10,000 mg/L [44]. However, in the case of propagation by air layering in guava (*Psidium guajava*), callus formation was reduced with increasing concentrations of IBA (2000, 4000, and 6000 mg/L), and its formation was completely inhibited at a concentration of 6000 mg/L IBA [45]. A similar trend was obtained in uva camarona (*Macleania rupestris*) propagation by air layering, where a reduction in the dry and fresh weights of the roots and callus was the result of concentrations greater than 1500 mg/L IBA [28]. We could argue that in air layering, the choice of the growth regulators and the concentration should be evaluated based on the crop, the genotype, and the growth and environmental features.

There was a great difference in the rooting rates and rooting characteristics among varieties of tree peony cultivars. This implied that there is significant research space for establishing efficient air-layering propagation protocols for various tree peony cultivars. The time cycle for sexual reproduction of tree peonies is too long, and thus asexual repro-

duction should be promoted in the future, particularly in the breeding of novel tree peony cultivars as well as ancient tree peonies. The current experiments considered the effects of the air-layering times as well as the growth regulator types and concentrations, but did not consider the effects of the wrapping matrix shape, the ring stripping incision, the growing media and substrate from the rhizosphere of this species, or microorganisms. These issues require further experimental investigation.

## 5. Conclusions

This study clearly demonstrated that the application of the air-layering technique in tree peonies could achieve successful propagation, and this has not been previously reported. It was suggested that air-layering time had a significant effect on the rooting rates among the three tree peony cultivars. The maximum rooting percentage (100%) was present in the 'Quehao' cultivar at the air-layering time of early summer (mid-June). The rooting parameters of 'Quehao' were negatively correlated with the concentration of NAA (1000–2000 mg/L), but positively correlated with IBA (1000–2000 mg/L), whereas those of 'Baoqing Hong' and 'Xishi' showed no dose dependence for the growth regulator concentration. The parameters of the root number, root tip number, and maximum root length of 'Quehao' were greater than those of the other two cultivars. The propagation of tree peonies with air layering could be an effective method for consistently obtaining plant material, especially for the promotion of high-quality tree peonies. Furthermore, this also provides an avenue for the study of the rooting mechanisms and nutritional breeding technologies of tree peonies.

**Author Contributions:** Y.Z. conducted the main experiments, collected and analyzed the data, and prepared the manuscript. Y.-H.H. and S.-Y.Y. conceived and revised the paper. All authors have read and agreed to the published version of the manuscript.

**Funding:** This research was funded by the Shanghai Science and Technology program (21DZ1202000), Bureau of Facility Support and Budget Chinese Academy of Sciences (ZSZY-001-8), Special Fund for Scientific Research of Shanghai Landscaping & City Appearance Administrative Bureau (G222412).

**Institutional Review Board Statement:** Not applicable.

**Informed Consent Statement:** Not applicable.

**Data Availability Statement:** All datasets generated during this study are included in the article.

**Acknowledgments:** We thank Jian-Feng Lv for his guidance on tree peony cultivation.

**Conflicts of Interest:** The authors declare that they have no conflict of interest.

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
