# Peer review of "Air Layering Improves Rooting in Tree Peony Cultivars from the Jiangnan Group"

_horticulturae, doi:10.3390/horticulturae8100941_

Round 1

Reviewer 1 Report

>Need to improve title of the manuscript. Very general and simple title need to improve my strong words.

>In abstract it mentioned that it has low propagation efficiency but didn't mention the traditional propagation method. 

>In abstract should mentioned about the complete methods of the study in which month experiment started with relation to phenological stage of the plant

>Material and methods, heading 2.2, authors should mention the season is it spring Autmn or summer, so reader can clearly understand about the season with respect to dates.

>Line 100 what is the branch meeting requirments?

>Line 105 why authors selected these concentration of 1000, 1500 and 2000 should refer with the review

>Line 118 formula should be inserted in formula format

>In Material and method or abstract, author didn't mention the experimental designs and factors.

>Figure 2. it's very clear and informative. But i have some suggestions for author's. Need to improve the figure quality. For bar color all the bar color should be the same for each graph for each treatment, i.e., IBA should be green or blue for all graphs. I don't understand about the stars of they are significant should mention the p-value and significant lettering if it is significant. 

>Table 2. Author should present p-value and lettering including the values of each parameter if they are significant or non-significant.

>Same suggestions for figure 3. 

>Conclusion should be clear, and I don't know about the journal format, conclusion should be in separate heading.  

Author Response

Reviewer reports:

Reviewer 1

Need to improve title of the manuscript. Very general and simple title need to improve my strong words.

--Reply: Thank you very much for your suggestion. We have improved the title of the manuscript (in red).

In abstract it mentioned that it has low propagation efficiency but didn't mention the traditional propagation method.

--Reply: We apologize for this confusion. We have rewritten the sentence to which you refer (“It is difficult to achieve large-scale production of excellent peony varieties because of its low propagation efficiency and long life cycle.”) in the revised manuscript (text in red in lines 9-11): It is difficult to achieve large-scale production of excellent peony varieties because of inefficient traditional propagation, including sowing, dividing, and grafting.

In abstract should mentioned about the complete methods of the study in which month experiment started with relation to phenological stage of the plant.

--Reply: Thank you. We have revised it in abstract in lines 12-15 (in red).

Material and methods, heading 2.2, authors should mention the season is it spring Autmn or summer, so reader can clearly understand about the season with respect to dates.

--Reply: Thank you for this suggestion. We have added the seasons spring and summer in lines 88-89 (in red) in the Material and methods heading 2.2.

Line 100 what is the branch meeting requirments?

--Reply: We apologize for this confusion. We have revised that in line 101 (in red) “five fresh branches of the current year were selected for air layering on each plant in mid-May, mid-June and mid-July.”

Line 105 why authors selected these concentration of 1000, 1500 and 2000 should refer with the review

--Reply: In combination with other studies on air layering [27-29] and our own experience in tree peony cultivation, we chose IBA and NAA for rooting hormone with concentration gradients of 0, 1000, 1500 and 2000 mg/L. We have a supplementary description in lines 109-111 in the revised manuscript and it is shown in red font.

Line 118 formula should be inserted in formula format.

--Reply: Thank you. We have inserted the formula in formula format in line 127 (in red).

In Material and method or abstract, author didn't mention the experimental designs and factors.

--Reply: We are sorry for that. We have added the experiment designs and factors in lines 12-15 in abstract (in red). We have rewrote the experiment design in lines 100-117 in materials and methods heading 2.3 and 2.4 (in red).

Figure 2. it's very clear and informative. But i have some suggestions for author's. Need to improve the figure quality. For bar color all the bar color should be the same for each graph for each treatment, i.e., IBA should be green or blue for all graphs. I don't understand about the stars of they are significant should mention the p-value and significant lettering if it is significant.

--Reply: Thank you for your good suggestion. Figure 2 has been modified in lines 201-212 in revised manuscript. We have colored all the bar the same color for each graph for each treatment. The a indicates significant differences, 0.01<p<0.05; b indicates highly significant differences, 0.01<p<0.05.

Table 2. Author should present p-value and lettering including the values of each parameter if they are significant or non-significant.

--Reply: Table 2 has been modified in lines 267-270 in revised manuscript (in red) and the p value has been added.

Same suggestions for figure 3.

--Reply: Figure 3 has been modified in lines 283-291 in revised manuscript (in red).

Conclusion should be clear, and I don't know about the journal format, conclusion should be in separate heading.

--Reply: We are sorry for missing the conclusion section and have added the conclusion in lines 365-378 in revised manuscript (in red).

Reviewer 2 Report

·         This area of research is interesting and scientifically sound good. The authors have written the manuscript very well. The manuscript is presented in a very simple form. The introduction is informative and sufficient. Methods and results are well described and the discussion section is justified with the obtained findings and with valid citation of relevant literature. But this article is not without its few drawbacks, which I am describing below:

·         Provide fully spelled-out terms before giving abbreviated terms in parentheses in the legends of tables, figures, as well as text, because the Materials and Methods section appeared later than the Results section.

·         Did the authors supply any amount of nutrients to the tree peony seeds?

·         Why were these three peony varieties selected?

·         Experiment design was not clear, and how many replicates per treatment and how many plants are in each replicate?

·         Three concentration gradients of NAA and IBA (1000 mg/L, 1500 mg/L, and 2000 mg/L) were applied to the ring-barked surfaces of three selected peony varieties, why did the authors select these three concentrations? Any selection criteria?

·         In the results section, there is a striking lack of connectors between sentences and leading to confusion. In this section, the author should only concentrate on the underlying results rather than mentioning other’s studies.

·         Discussion is very shallow and needs in-depth discussion with the recent literature published. In discussion, there is a lack of a mechanistic approach.

·         The manuscript was missing a conclusion!

·         There are many problems with the format and style, and please follow this journal format and style in the references!

Author Response

Reviewer reports:

Reviewer 2

This area of research is interesting and scientifically sound good. The authors have written the manuscript very well. The manuscript is presented in a very simple form. The introduction is informative and sufficient. Methods and results are well described and the discussion section is justified with the obtained findings and with valid citation of relevant literature. But this article is not without its few drawbacks, which I am describing below:

Provide fully spelled-out terms before giving abbreviated terms in parentheses in the legends of tables, figures, as well as text, because the Materials and Methods section appeared later than the Results section.

--Reply: Thank you very much for your encouraging words and for these useful comments. We have added fully spelled-out terms before giving abbreviated terms in where needed, e. g. Line 106 and Line 210.

Did the authors supply any amount of nutrients to the tree peony seeds?

--Reply: We are confused by this question. In this study, the tree peony seeds were not chosen for research.

Why were these three peony varieties selected?

--Reply: Three tree peony cultivars, ‘Baoqing Hong’, ‘Quhao’, and ‘Xishi’ belong to the traditional Jiangnan cultivar group, which have good growth adaptability in East China [7,30]. Selecting these three varieties of tree peonies to do the air-layering experiment can avoid the poor experimental data due to the bad growth. We have made a supplementary explanation in lines 293-296 in the discussion section.

Experiment design was not clear, and how many replicates per treatment and how many plants are in each replicate?

--Reply: Thank you for this helpful suggestion. We have rewrote the experiment design in lines 100-117 in materials and methods heading 2.3 and 2.4 (in red). The number of duplicate treatments and plants used in the experimental design were clearly stated in the revised manuscript.

Three concentration gradients of NAA and IBA (1000 mg/L, 1500 mg/L, and 2000 mg/L) were applied to the ring-barked surfaces of three selected peony varieties, why did the authors select these three concentrations? Any selection criteria?

--Reply: In combination with other studies on air layering [27-29] and our own experience in tree peony cultivation, we chose IBA and NAA for rooting hormone with concentration gradients of 0, 1000, 1500 and 2000 mg/L. We have a supplementary description in lines 109-111 in revised manuscript and it is shown in red font.

In the results section, there is a striking lack of connectors between sentences and leading to confusion. In this section, the author should only concentrate on the underlying results rather than mentioning other’s studies.

--Reply: We have to make it clear that in the results section, we only present the results of our own research and do not mention the results of other’s study.

Discussion is very shallow and needs in-depth discussion with the recent literature published. In discussion, there is a lack of a mechanistic approach.

--Reply: We are sorry for that. We reviewed some references related to air-layering, and rewrote the discussion section in red in revised manuscript.

The manuscript was missing a conclusion!

--Reply: We are sorry for missing the conclusion section and have added the conclusion in lines 365-378 in revised manuscript (in red).

There are many problems with the format and style, and please follow this journal format and style in the references!

--Reply: For the parts that do not meet the requirements of the journal format, we have made corresponding corrections according to the requirements of the journal format.

Reviewer 3 Report

 The research findings in the manuscript are interesting, but with little innovative approach. The authors described the influence of selected phytohormones on rooting, which is widely researched and described in many publications. However, the authors undertook to develop a method of reproduction of a species that is economically important and this is the greatest value of this publication.

Detailed comments:

Abstract

Line 8 - Latin name incomplete and no italics

Introduction

The manuscript lacks a clearly defined research goal. Please supplement.

Materials and Methods

Please state where the peat came from and what was the pH of the water solution that was accumulating.

Was a sterile medium used? How was the substrate secured against the development of plant pathogens?

Was the substrate used from the rhizosphere of the Peonia plant?

Probably a new approach would be to use the substrate from the rhizosphere of this species or microorganisms that are Peonia endophytes. It is worth considering this possibility in the future, increasing the chances of rooting with the use of microorganisms interacting with the tested plant.

Results

Figures 2 and 3 are illegible (low resolution), please improve their quality.

Discussion

The discussion chapter is disappointing. The research results have not been thoroughly discussed. I am asking for an extension of the discussion, especially since there are a number of publications related to similar research topics and a method that has been known for many years.

E.g:

Yeboah, J. , Branoh Banful, B. , Boateng, P. , Amoah, F. , Maalekuu, B. and Lowor, S. (2014) Rooting Response of Air-Layered Shea (Vitellaria paradoxa) Trees to Media and Hormonal Application under Two Different Climatic Conditions. American Journal of Plant Sciences5, 1212-1219. doi: 10.4236/ajps.2014.59134.

Carlton, J., & Moffler, M. (1978). Propagation of Mangroves by Air-layering. Environmental Conservation, 5(2), 147-150. doi:10.1017/S0376892900005658

DURAN-CASAS, SantiagoVELOZA SUAN, ClaraMAGNITSKIY, Stanislav  and  LANCHEROS, Héctor Orlando.Evaluation of uva camarona (Macleania rupestris Kunth A.C. Smith) propagation with air layering. Agron. colomb.[online]. 2013, vol.31, n.1, pp.18-26. ISSN 0120-9965.

Naithani, D.C., Nautiyal, A.R., Rana, D.K. and Mewar, D., Effect of Time of Air Layering, IBA Concentrations, Growing Media and their Interaction on the Rooting Behaviour of Pant Prabhat Guava (Psidium guajava L.) under Sub-Tropical Condition of Garhwal Himalaya, Int. J. Pure App. Biosci. 6(3): 169-180 (2018). doi: http://dx.doi.org/10.18782/2320-7051.6673 

Author Response

Reviewer reports:

Reviewer 3

The research findings in the manuscript are interesting, but with little innovative approach. The authors described the influence of selected phytohormones on rooting, which is widely researched and described in many publications. However, the authors undertook to develop a method of reproduction of a species that is economically important and this is the greatest value of this publication.

--Reply: Thank you for your encouraging words and for these useful comments. The innovation of this study is that tree peony can be propagated by air-layering technology, which has not been reported before. Phytohormones is a kind of root treatment agent for air-layering branches of tree peony, and the optimal use of phytohormone is determined by studying the effects of phytohormone kinds and concentration on the rooting parameters, so as to facilitate the popularization and application of this technology.

Detailed comments:

Abstract

Line 8 - Latin name incomplete and no italics

--Reply: We have corrected the latin name in line 8 and line 26 (in red).

Introduction

The manuscript lacks a clearly defined research goal. Please supplement.

--Reply: Thanks for the good suggestion. We have supplemented the research goal in lines 73-76 in Introductions section (in red).

Materials and Methods

Please state where the peat came from and what was the pH of the water solution that was accumulating.

--Reply: Thanks. We bought the peat of Klasmann-Deilmann 413 from the agency and have supplemented the brand and model of peat and the pH of the water solution in line 95 (in red).

Was a sterile medium used? How was the substrate secured against the development of plant pathogens?

--Reply: We do not consider it necessary to use sterile media because of the mother plants for air-layering exposed to nature. We also did not control the development of plant pathogens in the substrate. We believe that the air-layering is done in nature, unlike tissue culture in the laboratory, so there is no need to deliberately control the growth of microorganisms in the substrate, and perhaps some microorganisms promote root development and growth. The mother plant is cultivated and maintained in accordance with normal tree peony. Our results fully demonstrate that, the maximum rooting percentage (100%) was present in ‘Quehao’ cultivar at the air-layering time in mid-June.

Was the substrate used from the rhizosphere of the Peonia plant?

--Reply: We have stated that the filled soaking medium consisted of 50% moss and 50% peat (Klasmann-Deilmann 413) in lines 94-95. The used substrate is not from the rhizosphere of the Paeonia plant. We really appreciate your advice, we may consider using the substrate from the rhizosphere of the Paeonia plant to study the effect of medium on air-layering in the further.

Probably a new approach would be to use the substrate from the rhizosphere of this species or microorganisms that are Peonia endophytes. It is worth considering this possibility in the future, increasing the chances of rooting with the use of microorganisms interacting with the tested plant.

--Reply: Yes, you are right. We also thought that rhizosphere microorganisms could promote rooting of tree peony and carried out some studies in this field. Your proposal of using the substrate from the rhizosphere of the Paeonia plant really provides us a novel research idea. Once again, our thanks.

Results

Figures 2 and 3 are illegible (low resolution), please improve their quality.

--Reply: We are sorry for that. Figure 2 and 3 have been modified in revised manuscript.

Discussion

The discussion chapter is disappointing. The research results have not been thoroughly discussed. I am asking for an extension of the discussion, especially since there are a number of publications related to similar research topics and a method that has been known for many years.

E.g:

Yeboah, J. , Branoh Banful, B. , Boateng, P. , Amoah, F. , Maalekuu, B. and Lowor, S. (2014) Rooting Response of Air-Layered Shea (Vitellaria paradoxa) Trees to Media and Hormonal Application under Two Different Climatic Conditions. American Journal of Plant Sciences, 5, 1212-1219. doi:10.4236/ajps.2014.59134.

Carlton, J., & Moffler, M. (1978). Propagation of Mangroves by Air-layering. Environmental Conservation, 5(2), 147-150. doi:10.1017/S0376892900005658.

DURAN-CASAS, Santiago; VELOZA SUAN, Clara; MAGNITSKIY, Stanislav and LANCHEROS, Héctor Orlando. Evaluation of uva camarona (Macleania rupestris Kunth A.C. Smith) propagation with air layering. Agron. colomb. 2013, 31(1), 18-26. ISSN 0120-9965.

Naithani, D.C., Nautiyal, A.R., Rana, D.K. and Mewar, D., Effect of Time of Air Layering, IBA Concentrations, Growing Media and their Interaction on the Rooting Behaviour of Pant Prabhat Guava (Psidium guajava L.) under Sub-Tropical Condition of Garhwal Himalaya, Int. J. Pure App. Biosci. 6(3): 169-180 (2018). doi:http://dx.doi.org/10.18782/2320-7051.6673.

--Reply: We are sorry for disappointing of you. According to the references you listed and we reviewed some references related to air-layering, we rewrote the discussion section in red in revised manuscript.
